# Synthesis and Evaluation of the Antibacterial Activities of 13-Substituted Berberine Derivatives

**DOI:** 10.3390/antibiotics9070381

**Published:** 2020-07-06

**Authors:** Hamza Olleik, Taher Yacoub, Laurent Hoffer, Senankpon Martial Gnansounou, Kehna Benhaiem-Henry, Cendrine Nicoletti, Malika Mekhalfi, Valérie Pique, Josette Perrier, Akram Hijazi, Elias Baydoun, Josette Raymond, Philippe Piccerelle, Marc Maresca, Maxime Robin

**Affiliations:** 1Aix Marseille Univ, CNRS, Centrale Marseille, iSm2, 13397 Marseille, France; hamza.olleik@live.com (H.O.); cendrine.nicoletti@univ-amu.fr (C.N.); mekhalfimalika@yahoo.fr (M.M.); josette.perrier@univ-amu.fr (J.P.); 2Department of Biology, American University of Beirut, Beirut 1107-2020, Lebanon; eliasbay@aub.edu.lb; 3Aix Marseille Univ, CNRS, INSERM, Institut Paoli-Calmettes, CRCM, 13397 Marseille, France; taher.yacoub7@gmail.com (T.Y.); laurent.hoffer@inserm.fr (L.H.); 4Aix Marseille Univ, Avignon Université, CNRS, IRD, IMBE, 13397 Marseille, France; gnansounoumartial@yahoo.fr (S.M.G.); h.kehna@gmail.com (K.B.-H.); valerie.pique@univ-amu.fr (V.P.); philippe.piccerelle@univ-amu.fr (P.P.); 5Laboratoire d’études et de Recherches en Chimie Appliquée (LERCA), Université d’Abomey-Calavi (UAC), Cotonou 01 BP 2009, Benin; 6Doctoral School of Science and Technology, Research Platform for Environmental Science (PRASE), Lebanese University, Beirut 5, Lebanon; Akram.Hijazi@ul.edu.lb; 7Hôpital Cochin, Service de Bactériologie, Université Paris 5, 75014 Paris, France; josette.raymond@aphp.fr

**Keywords:** berberine derivatives, anti-bacterial agents, microbial sensitivity tests, structure-activity relationship, resistant strain, FtsZ protein

## Abstract

The biological activities of berberine, a natural plant molecule, are known to be affected by structural modifications, mostly at position 9 and/or 13. A series of new 13-substituted berberine derivatives were synthesized and evaluated in term of antimicrobial activity using various microorganisms associated to human diseases. Contrarily to the original molecule berberine, several derivatives were found strongly active in microbial sensitivity tests against *Mycobacterium*, *Candida albicans* and Gram-positive bacteria, including naïve or resistant *Bacillus cereus*, *Staphylococcus aureus* and *Streptococcus pyogenes* with minimal inhibitory concentration (MIC) of 3.12 to 6.25 µM. Among the various Gram-negative strains tested, berberine’s derivatives were only found active on *Helicobacter pylori* and *Vibrio alginolyticus* (MIC values of 1.5–3.12 µM). Cytotoxicity assays performed on human cells showed that the antimicrobial berberine derivatives caused low toxicity resulting in good therapeutic index values. In addition, a mechanistic approach demonstrated that, contrarily to already known berberine derivatives causing either membrane permeabilization, DNA fragmentation or interacting with FtsZ protein, active derivatives described in this study act through inhibition of the synthesis of peptidoglycan or RNA. Overall, this study shows that these new berberine derivatives can be considered as potent and safe anti-bacterial agents active on human pathogenic microorganisms, including ones resistant to conventional antibiotics.

## 1. Introduction

According to the World Health Organization (WHO) “we are heading towards a post-antibiotic era in which common infections and minor injuries can once again kill” [1]. Indeed, antibiotic resistance is rising dangerously worldwide, threatening our ability to cure common infectious diseases that become harder, and sometimes impossible, to treat. Antibiotic resistance is associated to higher medical costs, prolonged hospital stays, and increased mortality. In 2017, WHO published its first ever list of antibiotic-resistant “priority pathogens”—a catalogue of 12 families of bacteria that pose the greatest threat to human health. This list is divided into three categories, i.e., critical, high and medium priority groups, according to the urgency of need for new antibiotics. The critical priority group includes *Acinetobacter*, *Pseudomonas* and *Klebsiella* strains that are resistant to a large number of antibiotics, including carbapenems. The high priority group includes *Enterococcus*, *Staphylococcus*, *Helicobacter* and *Salmonella* strains that are becoming resistant to various antibiotics. Finally, the medium priority group includes resistant strains of *Streptococcus* and *Shigella*. Although it was not mentioned in the WHO list of priority pathogens due to the fact that tuberculosis is targeted by other dedicated programs, the rise in the number of resistant and multiresistant *Mycobacterium tuberculosis* is also alarming as it infects approximately 25% of the world population and causes millions deaths per year. In addition to bacterial infections, infections caused by fungi such as *Candida albicans* remain an important cause of death in hospitalized and immunocompromised or critical ill patients [1]. Similar to bacteria, an increase in the prevalence of *Candida albicans* strains resistant to antifungal drugs is observed worldwide [2]. In this context, there is an urgent need to find safe natural or synthetic molecules active against pathogenic microorganisms, including resistant ones. Berberine is a natural molecule present in some medicinal plants, mainly of the Berberidaceae family (*Berberis aquifolium*, *B. aristata*, *B. asiatica*, *B. chitria*, *B. lyceum*, *B. vulgaris*) but also in other species such as *Argemone mexicana*, *Coptis japonica*, *Hydrastis canadensis*, *Papaver dubium*, *Thalictrum flavum* [3]. Berberine and its derivatives [4] are known for having various biological activities such as antimalarial [5], antileishmanial [6], antiviral [7,8], hypolipidemic [9], anticancer [10,11] and antimicrobial effects [12,13,14]. Most of the active derivatives of berberine were obtained by substitution of various group at the position 9 and/or 13 resulting in higher activities compared to the parent molecule. For example, regarding antimicrobial activity, it has been shown that substitution at the 13th position by phenyl (A), ether link (B) or cycloberberine (C) (Figure 1) resulted in a higher activity against *Candida* species and *S. aureus* [14], including the methicillin-resistant strain MRSA [13].

Based on that, in the present study, a series of new derivatives were synthetized by substitution at position 13 and tested in term of their antimicrobial (antibacterial and antifungal) activity, innocuity on human cells and mechanism of action. The results showed that substitutions with polyphenol rings and chalcone moiety leaded to the more active and safer compounds. In addition, mechanistic studies revealed that antimicrobial activity of derivatives relies on their ability to inhibit the synthesis of cell wall and RNA. 

## 2. Materials and Methods

### 2.1. Chemistry

^1^H- and ^13^C-NMR spectra were recorded on an Avance III nanobay-300 MHz instrument (Bruker, Bremen, Germany, 300 MHz for ^1^H, 75 MHz for ^13^C). Chemical shifts are reported in ppm relative to the solvent in which the spectrum was recorded [^1^H: δ (d_6_-DMSO) = 2.50 ppm, δ (CDCl_3_) = 7.27 ppm; ^13^C: δ (d_6_-DMSO) = 39.52 ppm, δ (CDCl_3_) = 77.16 ppm]. Combustion analyses were performed at the analysis facilities of Spectropole (https://fr-chimie.univ-amu.fr/spectropole) with a Thermo Finnigan (San Jose, CA, USA) EA 1112 apparatus; all compounds had purity higher than 95%. Microwave-assisted reactions were performed in a CEM Discover microwave reactor with a focused field (CEM Corporation, Matthews, NC, USA). Silica gel F-254 plates (0.25 mm; Merck, Darmstadt, Germany) were used for thin-layer chromatography (TLC), and silica gel 60 (200–400 mesh; Merck) was used for flash chromatography. Unless otherwise stated, reagents were obtained from commercial sources and were used without further purification. Dihydroberberine (compound **1**), 13-hydroxyberberine (compound **2**), 8-acetonyl dihydroberberine (compound **3**) (yield: 85,40 and 85%) were synthesized by a procedure described in the literature [14].

#### 2.1.1. General Procedure for the Synthesis of Compounds **4**–**20**

Berberine chloride (3.71 g, 10 mmol) was dissolved in 5N NaOH (20 mL) under stirring at room temperature. Acetone (5 mL) was added dropwise at that temperature and stirred for 1 h, precipitation occurred during that time and the reaction mixture was filtered and washed with 80% MeOH to give the desired acetonylberberberine (compound **3**) (3.34 g, 85%). Compound **3** (1 g, 2.5 mmol) was used without further purification, dissolved in acetonitrile, NaI (0.5 g, 3.3 mmol) was added with various methylbromide (3 mmol) at 80 °C for 4 h. The reaction mixture was concentrated under vaccuo and chromatographed on silica gel (CH_2_Cl_2_/CH_3_OH, 90/10 *v*/*v*) to give compound **4**–**20**.

##### 13-(Acetic Acid Ethylester) Berberine (Compound **4**)

Yellow solid; yield: 26%; mp: 216 °C [12]; ^1^H-NMR (CDCl_3_) δ ppm: 10.56 (1H, s, H-8), 7.86 (1H, d, *J* = 9.2 Hz, H-12), 7.73 (1H, d, *J* = 9.2 Hz, H-11), 7.24 (1H, s, H-1), 6.90 (1H, s, H-4), 6.10 (2H, s, -OCH_2_O-), 5.20 (2H, brs, H-6), 4.37 (3H, s, OCH_3_-9), 4.35 (2H, q, *J* = 7.1 Hz, H-3′), 4.28 (2H, s, H-1′), 4.07 (3H, s, OCH_3_-10), 3.26 (2H, t, *J* = 5.1 Hz, H-5), 1.37 (3H, t, *J* = 7.1 Hz, H-4′).^13^C-RMN (CDCl_3_) δ ppm: 170.6 (C-2′), 150.7 (C-10), 150.2 (C-3), 147.5 (C-2), 147.2 (C-8), 146.5 (C-9), 137.6 (C-13a), 134.11 (C-4a), 133.42 (C-12a), 125.93 (C-12), 125.68 (C-13), 121.79 (C-8a), 119.79 (C-13b), 119.34 (C-11), 109.12 (C-4), 108.66 (C-1), 102.13 (-OCH_2_O-), 63.12 (OCH_3_-C9), 62.29 (C-3′), 57.41 (C-6), 56.97 (OCH_3_-C10), 37.15 (C-1′), 28.50 (C-5), 14.2 (C-4′). Anal. calcd. for C_24_H_24_BrNO_6_: C, 57.38; H, 4.82; N, 2.79. Found: C, 57.44; H, 4.87; N, 2.63.

##### 13-(Acetic Acid) Berberine (Compound **5**)

Yield 53%; ^1^H-RMN (CDCl_3_) δ ppm: 10.56 (1H, s, H-8), 7.86 (1H, d, *J* = 9.2 Hz, H-12), 7.73 (1H, d, *J* = 9.2 Hz, H-11), 7.24 (1H, s, H-1), 6.90 (1H, s, H-4), 6.10 (2H, s, -OCH_2_O-), 5.20 (2H, brs, H-6), 4.37 (3H, s, OCH_3_-9), 4.35 (2H, q, *J* = 7.1 Hz, H-3′), 4.28 (2H, s, H-1′), 4.07 (3H, s, OCH_3_-10), 3.26 (2H, t, *J* = 5.1 Hz, H-5), 1.37 (3H, t, *J* = 7.1 Hz, H-4′). ^13^C-RMN (CDCl_3_) δ ppm: 168.89 (COOH’), 150.66 (C-10), 149.21 (C-3), 147.46 (C-2), 147.21 (C-8), 145.31 (C-9), 137.60 (C-13a), 134.11 (C-4a), 133.11 (C-12a), 130.24 (C-13), 125.93 (C-12), 121.79 (C-8a), 119.79 (C-13b), 119.34 (C-11), 109.12 (C-4), 108.66 (C-1), 102.01 (-OCH_2_O-), 63.12 (OCH_3_-C9), 57.41 (C-6), 56.97 (OCH_3_-C10), 40.03 (C-1′), 28.50 (C-5). Anal. calcd. for C_22_H_20_BrNO_6_: C, 55.71; H, 4.25; N, 2.95. Found: C, 55.73; H, 4.32; N, 2.89.

##### 13-(4-Fluorobenzyl)Berberine (Compound **6**)

Yellow solid; yield: 55%; mp: 238 °C; ^1^H-RMN (CDCl_3_) δ ppm: 10.45 (1H, s, H-8), 7.73 (1H, d, *J* = 9.3 Hz, H-11), 7.60 (1H, d, *J* = 9.3 Hz, H-12), 7.08 (2H, m, H-4′), 7.08 (2H, m, H-3′), 6.92 (1H, s, H-1), 6.89 (1H, s, H-4), 6.02 (2H, s, -OCH_2_O-), 5.20 (2H, brs, H-6), 4.65 (2H, s, H-1′), 4.39 (3H, s, OCH_3_-9), 4.04 (3H, s, OCH_3_-10), 3.29 (2H, t, *J* = 5.7 Hz, H-5). ^13^C-RMN (CDCl_3_) δ ppm: 161.72 (d, *J_CF_* = 246.5 Hz, 1C, C-5′), 150.52 (C-10), 149.93 (C-3), 147.25 (C-8), 147.10 (C-2), 146.3 (C-9), 137.37 (C-13a), 133.81 (d, *J_CF_* = 3.3 Hz, 1C, C-2′), 133.81 (C-4a), 133.30 (C-12a), 129.61 (C-13), 129.40 (d, *J_CF_* = 7.8 Hz, 1C, C-3′), 125.63 (C-12), 121.91 (C-8a), 120.74 (C-11), 119.86 (C-13b), 116.35 (d, *J_CF_* = 21.5 Hz, 1C, C-4′), 108.64 (C-1), 108.64 (C-4), 101.96 (-OCH_2_O-), 63.07 (OCH_3_-C9), 57.50 (C-6), 56.86 (OCH_3_-C10), 35.68 (C-1′), 28.53 (C-5). Anal. calcd. for C_27_H_23_BrFNO_4_: C, 61.84; H, 4.42; N, 2.67. Found: C, 61.75; H, 4.46; N, 2.61.

##### 13-(4-Cyanobenzyl)Berberine (Compound **7**)

Dark green solid; yield: 60%; ^1^H-RMN (300 MHz, CDCl_3_) δ ppm: 10.49 (1H, s, H-8), 7.73 (1H, d, *J* = 9.2 Hz, H-11), 7.67 (1H, d, *J* = 8.4 Hz, H-4′), 7.51 (1H, d, *J* = 9.2 Hz, H-12), 7.31 (1H, d, *J* = 8.4 Hz, H-3′), 6.90 (1H, s, H-4), 6.77 (1H, s, H-1), 6.02 (2H, s, -OCH_2_O-), 5.20 (2H, brs, H-6), 4.65 (2H, s, H-1′), 4.35 (3H, s, OCH_3_-9), 4.04 (3H, s, OCH_3_-10), 3.30 (2H, t, *J* = 5.7 Hz, H-5). ^13^C-RMN (CDCl_3_) δ ppm: 150.64 (C-10), 150.20 (C-3), 147.29 (C-2), 147.20 (C-8), 146.38 (C-9), 143.77 (C-2′), 137.74 (C-13a), 133.76 (C-4a), 133.21 (C-12a), 133.21 (C-4′), 128.88 (C-3′), 128.46 (C-13), 125.87 (C-12), 121.96 (C-8a), 120.47 (C-11), 119.71 (C-13b), 118.29 (C-CN), 111.43 (C-5′), 108.76 * (C-4), 108.44 * (C-1),102.12 (-OCH_2_O-), 63.13 (OCH_3_-C-9), 57.86 (C-6), 56.89 (OCH_3_-C-10), 36.81 (C-1′), 28.48 (C-5). Anal. calcd. for C_28_H_23_BrN_2_O_4_: C, 63.29; H, 4.36; N, 5.27. Found: C, 63.02; H, 4.41; N, 5.22.

##### 13-(4-Iodomethylbenzyl)Berberine (Compound **8**)

Yellow solid; yield: 28%; ^1^H-RMN (300 MHz, CDCl_3_) δ ppm: 10.41 (1H, s, H-8), 7.72 (1H, d, *J* = 9.3 Hz, H-11), 7.59 (1H, d, *J* = 9.3 Hz, H-12), 7.38 (2H, d, *J* = 8.3 Hz, H-4′), 7.06 (2H, d, *J* = 8.3 Hz, H-3′), 6.91 (1H, s, H-1), 6.88 (1H, s, H-4), 6.00 (2H, s, -OCH_2_O), 5.20 (2H, brs, H-6), 4.64 (2H, s, H-1′), 4.45 (2H, s, H-6′), 4.41 (3H, s, OCH_3_-9), 4.03 (3H, s, OCH_3_-10), 3.30 (2H, t, *J* = 5.8 Hz, H-5). ^13^C-RMN (300 MHz, CDCl_3_) δ ppm: 150.46 (C-10), 150.03 (C-3), 147.19 (C-2), 146.49 (C-8), 146.30 (C-9), 138.34 (C-5′), 137.85 (C-2′), 137.58 (C-13a), 133.65 * (C-4a), 133.55 * (C-12a), 129.83 (C-4′), 129.81 (C-13), 128.38 (C-3′), 125.76 (C-12), 121.80(C-8a), 111.82 (C-11), 119.92 (C-13b), 108.79 (C-1), 108.57 (C-4), 63.21 (-OCH_2_O-), 63.21 (OCH_3_-C9), 58.13 (C-6), 56.83 (OCH_3_-C10), 36.36 (C-1′), 28.37 (C-5), 4.73 (C-6′). * may be reverse. Anal. calcd. for C_28_H_25_BrINO_4_: C, 52.03; H, 3.90; N, 2.17. Found: C, 51.93; H, 3.98; N, 2.09.

##### 13-(4-Ethenylbenzyl)Berberine (Compound **9**)

Yellow solid; yield: 18%; ^1^H-RMN (DMSO-d_6_) δ ppm: 10.02 (1H, s, H-8), 8.10 (1H, d, *J* = 9.4 Hz, H-11), 7.81 (1H, d, *J* = 9.4 Hz, H-12), 7.17 (1H, s, H-4), 7.03 (1H,s, H-1), 6.88 (1H, d, *J* = 7.9, H-7’), 6.83 (1H, brs, H-3′), 6.56 (1H, brd, *J* = 7.9, H-6′), 6.10 (2H, s, -OCH2O- [2,3]),), 6.02 (2H, s, -OCH2O- [4′,5′]), 4.88 (2H, brs, H-6), 4.64 (2H, s, H-1′), 4.12 (3H, s, OCH3-9), 4.03 (3H, s, OCH3-10), 3.16 (3H, t, *J* = 5.1 Hz, H-5). ^13^C-RMN (DMSO-d_6_) δ ppm: 150.17 (C-10), 149.17 (C-3), 147.90 (C-4′), 146.40 (C-2), 146.02 (C-5′), 145.40 (C8), 144.18 (C-9), 137.14 (C-13a),), 134.01 (C-4a), 132.83 (C-2′), 132.76 (C-12a), 130.11 (C-13), 126.12 (C-11), 121.67 (C-12), 121.29 (C-8a), 121.05 (C-7′), 120.02 (C-13b), 108.65 * (C-3′), 108.58 * (C-6′), 108.47 (C-4), 108.15 (C-1), 102.08 (-OCH2O- [2,3]), 101.17 (-OCH2O- [4′,5′]), 62.21 (OCH3-C-9), 56.98 (C-6), 56.93 (OCH3-C-10), 35.14 (C-1′), 27.24(C-5). * may be reverse. Anal. calcd. for C_29_H_26_BrNO_4_: C, 65.42; H, 4.92; N, 2.63. Found: C, 65.36; H, 5.01; N, 2.58.

##### 13-(4-Sulfamoylbenzyl)Berberine (Compound **10**)

Green solid; yield: 73%; ^1^H-RMN (DMSO-d_6_) δ ppm: 10.05 (1H, s, H-8), 8.08 (1H, d, *J* = 9.3 Hz, H-11), 7.80 (2H, d, *J* = 8.0Hz, H-4’), 7.73 (1H, d, *J* = 9.3 Hz, H-12), 7.38 (2H, d, *J* = 8.0Hz, H3’), 7.17 (1H, s, H-4), 6.89 (1H, s, H-1), 6.08 (2H, s, -OCH2O-), 4.89 (2H, s, H-6), 4.84 (2H, brs, H-1′), 4.13 (3H, s, OCH3-9), 4.02 (3H, s, OCH3-10), 3.18 (3H, t, J = 5.1 Hz, H-5). ^13^C-RMN (DMSO-d_6_) δ ppm: 150.22 (C-10), 149.25 (C-3), 146.40 (C-2), 145.61 (C8), 144.36 (C-9), 143.20 * (C-2′), 142.59 * (C-5′), 137.34 (C-13a), 134.10 (C4a), 132.57 (C12a), 128.57 (C-3′), 126.29 (C-11), 126.29 (C-4′), 121.47 (C-12), 121.30 (C-8a), 119.89 (C-13b), 108.51 (C-4), 108.00 (C-1), 102.07 (-OCH2O-), 62.06 (OCH3-C-9), 56.95 (C-6), 56.95 (OCH3-C-10), 35.41 (C-1′), 27.23 (C-5). * (C13) et (C-4″) not observed. Anal. calcd. for C_27_H_25_BrN_2_O_6_S: C, 55.39; H, 4.30; N, 4.78. Found: C, 55.33; H, 4.38; N, 4.88.

##### 13-(4-Aminomethylbenzyl)Berberine (Compound **11**)

Reddish solid; yield: 10%; ^1^H-RMN (CDCl_3_) δ ppm**:** 10.05 (1H, s, H-8), 8.17 (2H,brs, NH_2_), 8.06 (1H, d, *J* = 9.4 Hz, H-11), 7.72 (1H, d, *J* = 9.4 Hz, H-12), 7.45 (2H, d, *J* = 8.4 Hz, H-4′), 7.24 (2H, d, *J* = 8.4 Hz, H-3′), 6.93 (1H, s, H-1), 7.18 (1H, s, H-4), 6.08 (2H, s, -OCH_2_O), 4.88 (2H, brs, H-6), 4.76 (2H, brs, H-1′), 4.14 (3H, s, OCH_3_-9), 4.03 (2H, sl, H-6′), 4.03 (3H, s, OCH_3_-10), 3.17(2H, t, *J* = 5.8 Hz, H-5). ^13^C-RMN (CDCl_3_) δ ppm: 150.16 (C-10), 149.19 (C-3), 147.26 (C-2), 146.97 (C-8), 146.30 (C-9), 145.12 (C-2′), 137.69 (C-13a), 134.08 (C-5′), 133.69 (C-4a), 133.37 (C-12a), 129.53 (C-4′), 128.93 (C-13), 128.26 (C-3′), 125.74 (C-12), 121.90 (C-8a), 120.53 (C-11), 119.78 (C-13b), 108.69 * (C-4), 108.56 * (C-1), 102.06 (-OCH_2_O-), 62.05 (OCH_3_-C-9), 57.02 (C-6), 56.99 (OCH_3_-C-10), 41.85 (C-6′), 40.35 (C-1′), 27.21 (C-5). Anal. calcd. for C_28_H_27_BrN_2_O_4_: C, 62.81; H, 5.08; N, 5.23. Found: C, 62.78; H, 5.12; N, 5.24.

##### 13-(4-Formylbenzyl)Berberine (Compound **12**)

Red solid; yield: 75%; ^1^H-RMN (CDCl_3_) δ ppm: 10.02 (1H, s, H-CHO), 10.46 (1H, s, H-8), 7.90 (2H, d, *J* = 8.3 Hz, H-4′), 7.71 (1H, d, *J* = 9.3 Hz, H-11), 7.54 (1H, d, *J* = 9.3 Hz, H-12), 7.36 (2H, d, *J* = 8.3 Hz, H-3′), 6.90 (1H, s, H-4), 6.83 (1H, s, H-1), 6.01 (2H, s, -OCH_2_O-), 5.20 (2H, brs, H-6), 4.79 (2H, s, H-1′), 4.41 (3H, s, OCH_3_-9), 4.03 (3H, s, OCH_3_-10), 3.32 (2H, t, *J* = 5.5 Hz, H-5). ^13^C-RMN (CDCl_3_) δ ppm: 191.44 (C-CHO), 150.58 (C-10), 150.14 (C-3), 147.26 (C-2), 146.97 (C-8), 146.30 (C-9), 145.20 (C-2′), 137.69 (C-13a), 135.45 (C-5′), 133.69 (C-4a), 133.37 (C-12a), 130.78 (C-4′), 128.93 (C-13), 128.71 (C-3′), 125.74 (C-12), 121.90 (C-8a), 120.53 (C-11), 119.78 (C-13b), 108.69 * (C-4), 108.56 * (C-1), 102.06 (-OCH_2_O-), 63.16 (OCH_3_-C-9), 57.98 (C-6), 56.86 (OCH_3_-C-10), 36.91 (C-1′), 28.40 (C-5). Anal. calcd. for C_28_H_24_BrNO_5_: C, 62.93; H, 4.53; N, 2.62. Found: C, 62.84; H, 4.67; N, 2.60.

##### 13-[(2*H*-1,3-Benzodioxol-5-yl)methyl]Berberine (Compound **13**)

Yellow solid; yield: 40%; ^1^H-RMN (DMSO-d_6_) δ ppm: 10.02 (1H, s, H-8), 8.10 (1H, d, *J* = 9.4 Hz, H-11), 7.81 (1H, d, *J* = 9.4 Hz, H-12), 7.17 (1H, s, H-4), 7.03 (1H,s, H-1), 6.88 (1H, d, *J* = 7.9, H-7’), 6.83 (1H, brs, H-3′), 6.56 (1H, brd, *J* = 7.9, H-6′), 6.10 (2H, s, -OCH_2_O- [2,3]), 6.02 (2H, s, -OCH_2_O- [4′,5′]), 4.88 (2H, brs, H-6), 4.64 (2H, s, H-1′), 4.12 (3H, s, OCH_3_-9), 4.03 (3H, s, OCH_3_-10), 3.16 (3H, t, J = 5.1 Hz, H-5). ^13^C-RMN (DMSO-d_6_) δ ppm: 150.17 (C-10), 149.17 (C-3), 147.90 (C-4′), 146.40 (C-2), 146.02 (C-5′), 145.40 (C8), 144.18 (C-9), 137.14 (C-13a), 134.01 (C-4a), 132.83 (C-2′), 132.76 (C-12a), 130.11 (C-13), 126.12 (C-11), 121.67 (C-12), 121.29 (C-8a), 121.05 (C-7′), 120.02 (C-13b), 108.65 * (C-3′), 108.58 * (C-6′), 108.47 (C-4), 108.15 (C-1), 102.08 (-OCH_2_O- [2,3]), 101.17 (-OCH_2_O- [4′,5′]), 62.21 (OCH_3_-C-9), 56.98 (C-6), 56.93 (OCH_3_-C-10), 35.14 (C-1′), 27.24(C-5). * may be reverse. Anal. calcd. for C_29_H_26_BrNO_4_: C, 61.10; H, 4.40; N, 2.54. Found: C, 61.02; H, 4.48; N, 2.51.

##### 13[(2,3,4-Trimethoxyphenyl)methyl]Berberine (Compound **14**)

Brown solid; yield: 51%; ^1^H-RMN (DMSO-d_6_) δ ppm: 10.02 (1H, s, H-8), 8.12 (1H, d, *J* = 9.3 Hz, H-11), 7.77 (1H, d, *J* = 9.3 Hz, H-12), 7.16 (1H, s, H-4), 6.92 (1H, s, H-1), 6.65 (1H, d, *J* = 8.6 Hz, H-7’), 6.34 (1H, d, *J* = 8.6 Hz, H-6′), 6.08 (2H, s, -OCH2O-),4.88 (2H, brs, H-6), 4.51 (2H, brs, H-1′), 4.13 (3H, s, OCH3-9), 4.04 (3H, s, OCH3-10), 3.91 (3H, s, OCH3-5′), 3.76 * (3H, s, OCH3-3′), 3.75 * (3H, s, OCH3-4′), 3.16 (3H, t, *J* = 5.7 Hz, H-5). ^13^C-RMN (DMSO-d_6_) δ ppm: 152.64 (C-3′), 150.61 (C-5′), 150.12 (C-10), 149.13 (C-3), 146.38 (C-2), 145.21 (C8), 144.14 (C-9), 142.04 (C-4′), 137.03 (C-13a), 133.93 (C-4a), 132.94 (C-12a), 130.11 (C-13), 126.22 (C-11), 124.42 (C-2′), 122.72 (C-7′), 121.46 (C-12), 121.11 (C-8a), 119.98 (C-13b), 108.40 (C-1), 108.04 (C-4), 107.84 (C-6′), 102.02 (-OCH2O-),62.04 (OCH3-C-9), 60.43 * (OCH3-C-3′), 60.30 * (OCH3-C-4′), 56.91 (C-6), 56.91 (OCH3-C-10), 55.76 (OCH3-C-5′), 30.51 (C-1′), 27.19(C-5). * may be reverse. Anal. calcd. for C_30_H_30_BrNO_7_: C, 60.41; H, 5.07; N, 2.35. Found: C, 60.36; H, 5.10; N, 2.35.

##### 13[(3,4,5-Trimethoxyphenyl)methyl]Berberine (Compound **15**)

Brown solid; yield: 50%; ^1^H-RMN (DMSO-d_6_) δ ppm: 10.03 (1H, s, H-8), 8.13 (1H, d, *J* = 9.3 Hz, H-11), 7.88 (1H, d, *J* = 9.3 Hz, H-12), 7.17 * (1H, brs, H-4), 7.15 * (1H,brs, H-1), 6.47 (2H, s, H-3′, H-7’), 6.11 (2H, s, -OCH2O-),4.89 (2H, brs, H-6), 4.69 (2H, s, H-1′), 4.14 (3H, s, OCH3-9), 4.04 (3H, s, OCH3-10), 3.65 (9H, brs, OCH3-4′, OCH3-5′, OCH3-6′), 3.17 (2H, brt, *J* = 4.8 Hz, H-5). ^13^C-RMN (DMSO-d_6_) δ ppm: 153.22 (C-4′), 153.22 (C-6′), 150.05 (C-10), 149.12 (C-3), 146.38 (C-2), 145.35 (C8), 144.08 (C-9), 137.14 (C-13a), 136.27 (C-5′), 134.68 (C-2′), 133.91 (C-4a), 132.86 (C-12a), 130.07 (C-13), 126.05 (C-11), 121.54 (C-12), 121.22 (C-8a), 120.08 (C-13b), 108.35 * (C-1), 108.33 * (C-4), 105.69 (C-3′), 105.69 (C-7′), 101.99 (-OCH2O-),61.97 (OCH3-C-9), 59.96 (OCH3-C-5′), 56.93 (C-6), 56.87 (OCH3-C-10), 56.10 (OCH3-C-4′), 56.10 (OCH3-C-6′), 35.55 (C-1′), 27.23(C-5).). * may be reverse. Anal. calcd. for C_30_H_30_BrNO_7_: C, 60.41; H, 5.07; N, 2.35. Found: C, 60.21; H, 5.02; N, 2.33.

#### 4-[2-(Berberin-13-yl) acetyl]-2,6-dimethoxyphenyl Benzoate (Compound **16**)

Yellow solid; yield: 16%; ^1^H-RMN (CDCl_3_) δ ppm: 10.03 (1H, s, H-8), 8.19(1H, d, *J* = 9.5 Hz, H-11), 8.15 (2H, m, H-3″), 7.82 (1H, d, *J* = 9.5 Hz, H-12), 7.80 (1H, m, H-5″), 7.65 (2H, m, H-4 ″), 7.64 (2H, s, H-4’), 7.19 (1H, s, H-4), 6.89 (1H, s, H-1), 6.11 (2H, s, -OCH2O-), 5.37 (2H, brs, H-6), 4.90 (2H, br s, H-1′), 4.14 (3H, s, OCH3-9), 4.07 (3H, s, OCH3-10), 3.90 (3H, s, OCH3-5’), 3.15 (3H, t, *J* = 5.6 Hz, H-5). ^13^C-RMN (CDCl_3_) δ ppm: 196.91 (C-2’), 163.24 (C-1″), 152.24 (C-5′), 153.31 (C-10), 149.44 (C-3), 146.79 (C-2), 145.69 (C-8), 144.20 (C-9), 134.35 (C-4a), 133.99 (C-5″), 133.59 (C-12a), 132.86 * (C-6′), 132.84 * (C-3′), 129.90 (C-3″), 129.14 (C-4″), 128.02 (C-13), 127.66 (C-2″), 126.28 (C-11), 121.52 (C-12), 120.90 (C-8a), 119.93 (C-13b), 108.52 (C-4), 107.91 (C-1), 105.74 (C-4’), 102.13 (-OCH2O-), 62.07 (OCH3-C-9), 57.02 (OCH3-C-10), 56.78 (C-6), 56.50 (OCH3-C-5’), 27.24 (C-5). * may be reverse. Anal. calcd. for C_37_H_32_BrNO_9_: C, 62.19; H, 4.51; N, 1.96. Found: C, 62.20; H, 4.48; N, 2.00.

#### 4-[2-(Berberin-13-yl)acetyl]-3-(benzoyloxy)phenyl Benzoate (Compound **17**)

Yellow solid; yield: 55%; ^1^H-RMN (DMSO-d_6_) δ ppm: 9.97 (1H, s, H-8), 8.52 (1H, d, *J* = 9.4 Hz, H-4’), 8.03 (4H, m, H-3″), 8.20 (4H, m, H-3″), 8.13(1H, d, *J* = 9.4 Hz, H-11), 7.89 (1H, d, *J* = 9.4 Hz, H-12), 7.80 (2H, m, H-5″), 7.66 (4H, m, H-4″), 7.62 (1H, brs, H-7′), 7.61 (1H, dd, *J* = 9.4 Hz, *J* = 2.0 Hz, H-5′), 7.15 (1H, s, H-4), 6.86 (1H, s, H-1), 6.16 (2H, s, -OCH_2_O-), 5.25 (2H, s, H-1′), 4.82 (2H, brs, H-6), 4.10 (3H, s, OCH_3_-9), 4.06 (3H, s, OCH_3_-10), 3.08 (3H, t, *J* = 4.9 Hz, H-5). ^13^C-RMN (DMSO-d_6_) δ ppm: 196.23 (C-2’), 164.18, 163.98 (C-1″), 154.74 (C-6′), 150.27 (C-10), 149.44 (C-3), 146.91 (C-2), 145.56 (C-8), 144.13 (C-9), 137.06 (C-13a), 134.43 (C-4a), 134.10, 134.07 (C-5″), 132.77 (C-12a), 132.19 (C-4′), 129.98,129.76 (C-4″), 129.05, 128.85 (C-3″), 128.42, 128.34 (C-2″), 127.08 (C-13), 126.97 (C-3′), 126.09 (C-11), 121.52 (C-12), 120.87 (C-8a), 120.40 (C-5′), 119.78 (C13b), 118.62 (C-7′), 108.50 (C-1), 107.89 (C-4), 102.20 (-OCH_2_O-), 62.02 (OCH_3_-C-9), 57.04(OCH_3_-C-10), 56.67 (C-6), 43.88 (C-1′), 27.24 (C-5). Anal. calcd. for C_42_H_32_BrNO_9_: C, 65.12; H, 4.16; N, 1.81. Found: C, 65.10; H, 4.21; N, 1.80.

#### 8-{4-[(Berberin-13-yl)methyl]phenoxy}-2*H*-chromen-2-one (Compound **18**)

Yellow solid; yield: 12%; ^1^H-RMN (DMSO-d_6_) δ ppm: 10.06 (1H, s, H-8), 8.09 (1H, d, *J* = 9.5 Hz, H-12), 8.00 (1H, d, *J* = 9.5Hz, H3″), 7.78(1H, d, *J* = 9.5 Hz, H-11), 7.65 (1H, d, *J* = 8.8Hz, H-4″), 7.47 (2H, d, *J* = 8.3Hz, H3’), 7.21 (2H, d, *J* = 8.3Hz, H-4’), 7.17 (1H, s, H-4), 7.08 (1H, d, *J* = 2.5, H-7″), 7.02 (1H, dd, *J* = 2.5Hz, *J* = 8.8Hz, H-5″), 6.95 (1H, s, H-1), 6.30 (1H, d, *J* = 9.5, H-2″), 6.07 (2H, s, -OCH2O-), 5.20 (1H, s, H-6’), 4.88 (2H, brs, H-6), 4.76 (2H, s, H-1′), 4.12 (3H, s, OCH3-9), 4.02 (3H, s, OCH3-10), 3.16 (3H, t, *J* = 5.5 Hz, H-5). ^13^C-RMN (DMSO-d_6_) δ ppm: 161.44 (C-6″), 160.26 (C-1″), 155.32 (C-8″), 152.22 (C-10), 149.21 (C-3), 146.39 (C-2), 145.50 (C-8), 144.31 (C-3″),144.23 (C-9), 139.12 (C-2’), 137.17 (C-13a), 137.76 (C-5′), 134.06 (C-4a), 132.69 (C-12a), 129.89 (C-13), 129.89 (C-4″), 129.54 (C-3’), 128.84 (C-4’), 126.17 (C-11), 121.67 (C-12), 121.26 (C-8a), 119.98 (C-13b), 112.96 (C-5″), 112.63 (C-2″), 112.53 (C-9″), 108.51 (C-4), 108.08 (C-1), 102.06 (-OCH2O-),101.56 (C-7″), 69.62 (C-6’), 62.07 (OCH3-C-9), 56.95 (C-6), 56.90 (OCH3-C-10), 35.28 (C-1’), 27,26 (C-5). Anal. calcd. for C_37_H_30_BrNO_7_: C, 65.30; H, 4.44; N, 2.06. Found: C, 65.25; H, 4.48; N, 2.02.

#### 9-{[4-(1,3-Benzothiazol-2-yl)phenyl]methyl}berberine (Compound **19**)

Red solid; yield: 20%; ^1^H-RMN (DMSO-d_6_) δ ppm: 10.06 (1H, s, H-8), 8.13 (1H, brd, *J* = 7.8, H-7″), 8.11 (1H, d*, J* = 9.4 Hz, H-11), 8.09 (2H, d, *J* = 8.1 Hz, H-4’), 8.04 (1H, brd, *J* = 7.8 Hz, H-4″), 7.81 (1H, d, *J* = 9.4 Hz, H-12), 7.55 (1H, brt, *J* = 7.8 Hz, H-5″), 7.47(1H, brt, *J* = 7.8Hz, H-6″), 7.39 (2H, d, *J* = 8.1 Hz, H3’), 7.18 (1H, s, H-4), 6.99 (1H, s, H-1), 6.08 (2H, s, -OCH2O-), 4.90 (2H, brs, H-6), 4.86 (2H, s, H-1′), 4.14 (3H, s, OCH3-9), 4.03 (3H, s, OCH3-10), 3.19 (3H, t, J = 5.3 Hz, H-5). ^13^C-RMN (DMSO-d_6_) δ ppm: 166.70 (C-2″), 153.50 (C-3a″), 150.19 (C-10), 149.23 (C-3), 146.40 (C-2), 145.54 (C8), 144.29 (C-9), 143.73 (C-2′), 137.27 (C-13a), 134.40 (C-7a″), 134.10 (C4a), 132.63 (C12a), 131.43 (C-5′), 129.45 (C-13), 129.01 (C-3′), 127.81 (C-4′),126.62(C-5″), 126.25 (C-11), 125.51 (C-6″), 122.77 (C-4″), 122.29 (C-7″), 121.54 (C-12), 121.22 (C-8a), 119.88 (C-13b), 108.46 (C-1), 108.11 (C-4), 102.02 (-OCH2O-),62.01 (OCH3-C-9), 56.89 (C-6), 56.89 (OCH3-C-10), 35.49 (C-1′), 27.22 (C-5). Anal. calcd. for C_34_H_27_BrN_2_O_4_S: C, 63.85; H, 4.26; N, 4.38. Found: C, 63.75; H, 4.12; N, 4.40.

#### 9-{[4-(1,3,4-Oxadiazol-2-yl)phenyl]methyl}berberine (Compound **20**)

Yellow solid; yield: 57%; ^1^H-RMN (DMSO-d_6_) δ ppm: 10.06 (1H, s, H-8), 9.33 (1H, s, H-2″), 8.09 (1H, d, *J* = 9.5 Hz, H-11), 8.02(2H, d, *J* = 8.3 Hz, H4′), 7.79 (1H, d, *J* =9.5 Hz, H-12), 7.42(2H, *J* = 8.3 Hz, H3′), 7.18 (1H, s, H-4), 6.94 (1H,s, H-1), 6.08 (2H, s, -OCH_2_O- [2,3]), 4.90 (2H, brs, H-6), 4.87 (2H, brs, H-1′), 4.14 (3H, s, OCH_3_-9), 4.03 (3H, s, OCH_3_-10), 3.18 (2H, t, *J* = 5.2 Hz, H-5). ^13^C-RMN (DMSO-d_6_) δ ppm: 163.35 (C-1″), 154.47 (C-2″), 150.22 (C-10), 149.3 (C-3), 146.5 (C-2), 145.61 (C8), 144.35 (C-9), 143.52 (C-2′), 137.92 (C-5′), 137.34 (C-13a), 134.11 (C-4a), 132.63 (C-12a), 129.34 (C-13), 129.11 (C-3′), 127.40 (C-4′), 126.28 (C-11), 121.79 (C-12), 121.51 (C-8a), 119.89 (C-13b), 108.50 (C-4), 108.11 (C-1), 102.07 (-OCH_2_O- [2,3]), 62.55 (OCH_3_-C-9), 57.44 (C-6), 57.44 (OCH_3_-C-10), 36.08 (C-1′), 27.27(C-5). Anal. calcd. for C_29_H_24_BrN_3_O_5_: C, 60.64; H, 4.21; N, 7.32. Found: C, 60.51; H, 4.25; N, 7.29.

#### N-({4-[(Berberin-13-yl)methyl]phenyl}methylidene)hydroxylamine (Compound **21**)

Compound **12** (0.4 g, 0.74 mmol), hydroxylamine hydrochloride (0.16 g, 2.2 mmol, 3 eq) in EtOH/H_2_O (10/2 mL) was added in sodium acetate (0.24 g, 3 mmol, 4 eq). The mixture was stirred at 60 °C for 1 h, poured onto water (50 mL). The solution was extracted with ether (3 × 20 mL) and the organic phase was removed under vacuum and reprecipitate in CHCl_3_. The yellow precipitate was collected washed with CHCl_3_ to give pure compound **21**. Yellow solid; yield: 10%; ^1^H-RMN (DMSO-d_6_) δ ppm: 11.21 (1H, s, OH), 10.02 (1H, s, H-8), 8.09 (1H, d, *J* = 9.4 Hz, H-11), 7.78 (1H, d, *J* = 9.4 Hz, H-12), 8.12 (1H, s, H-6′), 7.57 (2H, d, *J* = 8.1 Hz, H-4′),7.21 (2H, d, *J* = 8.1 Hz, H-3′), 7.16 (1H, s, H-1), 6.96 (1H, s, H-4), 6.08 (2H, s, -OCH_2_O-), 4.88 (2H, brs, H-6), 4.76 (2H, s, H-1′), 4.13 (3H, s, OCH_3_-9), 4.02 (3H, s, OCH_3_-10), 3.17 (2H, t, *J* = 5.7 Hz, H-5). ^13^C-RMN (DMSO-d_6_) δ ppm: 150.19 (C-10), 149.22 (C-3), 147.65 (C-6′), 146.40 (C-2), 145.49 (C-8), 144.30(C-9), 140.32 (C-2′), 137.22 (C-13a), 134.04 (C-5′), 133.12 (C-4a), 131.67 (C-12a), 129.68 (C-13), 128.41 (C-4′), 127.04 (C-3′), 126.21 (C-12), 121.59 (C-11), 121.27 (C-8a), 119.97 (C-13b), 108.49 (C-4), 108.15 (C-1), 102.05 (-OCH_2_O-), 62.05 (OCH_3_-C-9), 57.00 (OCH_3_-C-10) 56.93 (C-6), 35.43 (C-1′), 27.25 (C-5). Anal. calcd. for C_27_H_25_BrN_2_O_5_: C, 60.34; H, 4.69; N, 5.21. Found: C, 60.29; H, 4.71; N, 5.23.

#### (2*E*)-3-{4-[(Berberin-13-yl)methyl]phenyl}prop-2-enoic Acid (Compound **22**)

Compound **12** (1 g, 1.8 mmol), malonic acid (0.4 g, 4.1 mmol, 2.2 eq) in MeOH/ethyl acetate (5/2 mL) was added in acetic acid (15 mL). Then was added piperidine (0.5 mL, 4 mmol, 2.7 eq) and the reaction mixture was submitted to microwave irradiation in a CEM microwave apparatus (Power time, Acetic acid, Tmax: 110 °C, 30 min). The solution was poured onto ice/water (200 mL) and the precipitate was collected by sucion and washed with excess water to give **22** as a reddish powder. Red solid; yield: 50%; ^1^H-RMN (DMSO-d_6_) δ ppm: 10.04 (1H, s, H-8), 8.09 (1H, d, *J* = 9.4 Hz, H-11), 7.76 (1H, d, *J* = 9.4 Hz, H-12), 7.68 (2H, d, *J* = 8.2 Hz, H-4′), 7.58 (1H, d, *J* = 16.0 Hz, H-6′), 7.22 (2H, d, *J* = 8.2 Hz, H-3′), 7.17 (1H, s, H-1), 6.93 (1H, s, H-4), 6.53 (1H, d, *J* = 16.0 Hz, H-7′), 6.08 (2H, s, -OCH_2_O-), 4.87 (2H, brs, H-6), 4.77 (2H, s, H-1′), 4.12 (3H, s, OCH_3_-9), 4.02 (3H, s, OCH_3_-10), 3.16 (2H, t, *J* = 5.7 Hz, H-5). ^13^C-RMN (DMSO-d_6_) δ ppm: 167.78 (C-8′), 150.18 (C-10), 149.19 (C-3), 146.38 (C-2), 145.49 (C-8), 144.26 (C-9), 142.39 (C-6′), 141.13 (C-2′), 137.20 (C-13a), 134.03 (C-5′), 133.12 (C-4a), 132.65 (C-12a), 129.68 (C-13), 128.71 (C-4′), 128.55 (C-3′), 126.18 (C-12), 121.58 (C-11), 121.24 (C-8a), 120.40 (C-7′), 119.95 (C-13b),), 108.47 (C-4), 108.12 (C-1), 102.04 (-OCH_2_O-), 62.04 (OCH_3_-C-9), 56.95 (C-6), 56.90 (OCH_3_-C-10), 27.24(C-5). Anal. calcd. for C_30_H_26_BrNO_6_: C, 62.51; H, 4.55; N, 2.43. Found: C, 62.41; H, 4.58; N, 2.47.

#### (2*E*)-3-{4-[(Berberin-13-yl)methyl]phenyl}-N-[2-(4-sulfamoylphenyl)ethyl]prop-2-enamide (Compound **23**)

Compound **22** (0.2 g, 0.36 mmol), 1-(3-dimethyl-aminopropyl)-3-ethylcarbodiimide hydrochloride: EDCI (0.06 g, 0,4 mmol, 1,1 eq) and 1-hydroxybenzotriazole: HOBT (0.05 g, 0.4 mmol, 1.1 eq) in DMF (10 mL) were stirred for 2.5 h at r.t., then was added 4-(2-aminoethyl) benzene-sulfonamide (0.1 g, 0.45 mmol, 1.3 eq) under stirring for 12 h. The reaction mixture was poured onto ice/water (100 mL), the precipitate was filtered off and washed with excess water to give **23** as a dark green powder. Green solid; yield: 16%; ^1^H-RMN (DMSO-d_6_) δ ppm: 10.04 (1H, s, H-8), 8.21 (1H, t, *J* = 5.7 Hz, NH), 8.09 (1H, d, *J* = 9.3 Hz, H-11), 7.77 (1H, d, *J* = 9.3 Hz, H-12), 7.73 (2H, d, *J* = 8.2 Hz, H5″), 7.55 (2H, d, *J* = 8.0 Hz, H-4′), 7.41 (2H, d, *J* = 8.2 Hz, H-4″), 7.40 (1H, d, *J* = 15.8 Hz, H-6′), 7.30 (2H, s, NH_2_), 7.21 (2H, d, *J* = 8.0 Hz, H-3′), 7.17 (1H, s, H-4), 6.94 (1H, s, H-1), 6.57 (1H, d, *J* = 15.8 Hz, H-7′), 6.08 (2H, s, -OCH_2_O-), 4.87 (2H, brs, H-6), 4.76 (2H, s, H-1′), 4.12 (3H, s, OCH_3_-9), 4.02 (3H, s, OCH_3_-10), 3.44 (2H, m, H-1″), 3.16 (2H, t, *J* = 5.7 Hz, H-5), 2.85 (2H,t, *J* = 7.0 Hz, H-2″). ^13^C-RMN (DMSO-d_6_) δ ppm: 164.9 (C-8′), 150.2 (C-10), 149.2 (C-3), 146.4 (C-2), 145.5 (C-8), 144.2 (C-9), 143.6 (C-6′’), 142.0 (C-2′), 140.54 (C-6′), 138.0 (C-3″), 137.2 (C-13a), 134.0 (C-4a), 133.5 (C5′), 132.7 (C-12a), 129.7 (C-13), 129.1 (C-4″), 128.6 (C-3′), 128.2 (C-4′), 126.2 (C-11), 125.6 (C5″), 122.2 (C-7′), 121.6 (C-12), 121.2 (C-8a), 119.9 (C-13b), 108.5 (C-4), 108.1 (C-1), 102.3 (-OCH_2_O-), 62.0 (OCH_3_-C-9), 56.9 (C-6), 56.9 (OCH_3_-C-10), 35.4 (C-1′), 34.7 (C-2″), 27.2 (C-5). Anal. calcd. for C_38_H_36_BrN_3_O_7_S: C, 60.16; H, 4.78; N, 5.54. Found: C, 60.21; H, 4.81; N, 5.44.

#### N-{3-[(2*E*)-3-{4-[(Berberin-13-yl)methyl]phenyl}prop-2-enoyl]-4-hydroxyphenyl}acetamide (Compound **24**)

Compound **12** (0.44 g, 0.8 mmol, 0.7 eq) and N-(3-acetyl-4-hydroxyphenyl)acetamide (0.2 g, 1.1 mmol, 1.2 eq) were solubilized in methanol (20 mL). To this solution was added LiOH (0.15 g, 6 mmol, 8 eq.) and the solution was submitted to microwave irradiation using a CEM apparatus (mode open vessel, new method, mode standard, hold time: 2 min., power: 300 W, run time: 20 min.). At the end of the irradiation the solvent was removed under vacuum to give a red mixture. Addition of 1N HCl (50 mL) under ice give a yellow precipitate that was filtered with ether to give a yellow solid of **24**. Yellow solid; yield: 40%; ^1^H-RMN (DMSO-d_6_) δ ppm: 11,62 (1H, s, OH-2″), 10.05 (1H, s, H-8), 9.92 (1H, s, NH), 8.18 (1H, d, *J* = 2.5 Hz, H-6″), 8.10 (1H, d, *J* = 9.4 Hz, H-11), 7.80 (2H, d, *J* = 7.2 Hz, H-3′), 7.79 (1H, d, *J* = 9.4 Hz, H-12), 7.77 (1H, d, *J* = 15.6 Hz, H-6′), 7.73 (1H, d, *J* = 15.6 Hz, H-7′), 7.65 (1H, dd, *J* = 8.8 Hz, *J* = 2.5 Hz, H-4″), 7.28 (2H, d, *J* = 7.2 Hz, H-4′), 7.18 (1H, s, H-4), 6.96 (1H, s, H-1), 6.96 (1H, d, *J* = 8.8 Hz, H-3″), 6.09 (2H, s, -OCH_2_O-), 4.88 (2H, brs, H-6), 4.81 (2H, s, H-1′), 4.13 (3H, s, OCH_3_-9), 4.03 (3H, s, OCH_3_-10), 3.17 (2H, t, J = 5.4 Hz, H-5), 2.02 (3H,s, CH_3_-NHCOCH_3_). ^13^C-RMN (DMSO-d_6_) δ ppm: 193.2 (C-8′),168.2 (CO-NHCOCH_3_), 156.9 (C-2″), 150.6 (C-10), 149.5 (C-3), 146.2 (C-2), 145.2 (C-8), 144.5 (C-9), 143.6 (C-6′), 142.5 (C-2′), 142.4 (C-5′), 137.5 (C-13a), 134.3 (C-4a), 132.9 (C-12a), 129.9 (C-13), 129.1 (C-3′), 128.4 (C-4′), 127.6 (C-4″), 125.9 (C-11), 122.7 (C-7′), 121.6 (C-8a),121.6 (C-1″), 121.4 (C-5″), 121.3 (C-12), 120.6 (C-6″), 120.2 (C-13b), 117.4 (C-3″), 108.2 (C-4), 107.9 (C-1), 101.8 (-OCH_2_O-), 61.7 (OCH_3_-C-9), 56.6 (C-6), 56.6 (OCH_3_-C-10), 35.2 (C-1′), 27.0 (C-5), 23.4 (C-CH_3_ NHCOCH_3_). Anal. calcd. for C_38_H_33_BrN_2_O_7_: C, 64.32; H, 4.69; N, 3.95. Found: C, 64.30; H, 4.671; N, 4.00.

#### (2*E*)-3-{4-[(Berberin-13-yl)methyl]phenyl}-1-(2-hydroxy-4-methoxyphenyl)prop-2-en-1-one (Compound **25**)

Using the same protocol as for **24** with 2-hydroxy-4-methoxyacetophenone gives **25**. Green solid; yield: 78%; ^1^H-RMN (DMSO-d_6_) δ ppm: 10.08 (1H, s, H-8), 8.27 (1H, d, *J* = 9.1 Hz, H-6″), 8.10 (1H, d, *J* = 9.4 Hz, H-11), 8.01 (2H, d, *J* = 15.5 Hz, H-a),7.92 (2H, d, *J* = 8.3Hz, H-4’), 7.80 (2H, d, *J* = 15.5 Hz, H-b), 7.78 (1H, d, *J* = 9.4 Hz, H-12), 7.28 (2H, d, *J* = 8.3Hz, H3’), 7.18 (1H, s, H-4), 6.96 (1H, s, H-1), 6.56 (1H, dd, *J* = 2.5Hz, *J* = 9.1Hz, H-5″), 6.53 (1H, d, *J* =2.5, H-3″), 6.08 (2H, s, -OCH2O-), 4.90 (2H, brs, H-1′), 4.81 (2H, brs, H-6), 4.13 (3H, s, OCH3-9), 4.02 (3H, s, OCH3-10), 3.85 (3H, s, OCH3-4″),3.17 (3H, t, *J* =5.7 Hz, H-5). ^13^C-RMN (DMSO-d_6_) δ ppm: 191.82 (C-b’), 166.04 (C-2″), 165.68 (C-4″), 150.24 (C-10), 149.28 (C-3), 146.46 (C-2), 145.59 (C-8), 144.30 (C-9), 143.52 (C-b, 142.19 (C-2’), 137.25 (C-13a), 134.10 (C-4a), 133.59 (C-5′), 132.78 (C-6″), 132,66 (C-12a), 129.83 (C-4’), 129.63 (C-13), 128.67 (C-3’), 126.23 (C-11), 121.64 (C-12), 121,29 (C-8a), 121.26 (C-a), 119,97 (C-13b), 113.91 (C-1″), 108.54 (C-4), 108.14 (C-1), 107.49 (C-5″), 102.11 (-OCH2O-), 100.95 (C-3″),62.10 (OCH3-C-9), 56,98 (C-6), 56,93 (OCH3-C-10), 55.81 (OCH3-4″), 35.59 (C-1’), 27,29 (C-5). Anal. calcd. for C_37_H_32_BrNO_7_: C, 65.11; H, 4.73; N, 2.05. Found: C, 65.02; H, 4.76; N, 2.01.

#### 2-{4-[(Berberin-13-yl)methyl]phenyl}-7-methoxy-4H-chromen-4-one (Compound **26**)

Compound **25** (0.17 g, 0.24 mmol) was solubilized in a solution of DMSO (20 mL) and Iodine (17 mg, 10% weight). The solution was submitted to microwave irradiation (CEM, Power time, Tmax: 140 °C, 30 min). The solution was poured on ice/1N HCl (100 mL) to give a yellow precipitate of **26** which was washed with excess water. Yellow solid; yield: 18%; ^1^H-RMN (DMSO-d_6_) δ ppm: 10.06 (1H, s, H-8), 8.10 (1H, d, *J* = 9.3 Hz, H-11), 8.06 (2H, d, *J* = 8.3Hz, H-4’), 7.95 (1H, d, *J* = 8.5Hz, H-5″), 7.79 (1H, d, *J* = 9.3 Hz, H-12), 7.38 (2H, d, *J* = 8.3Hz, H3’), 7.29 (1H, d, *J* = 2.2Hz, H8″), 7.18 (1H, s, H-4), 7.08 (1H, dd, *J* = 8.5Hz, *J* = 2.2Hz, H-6″), 6.97 (1H, s, H3″), 6.95 (1H, s, H-1), 6.08 (2H, s, -OCH2O-), 4.90 (2H, brs, H-6), 4.85 (2H, s, H-1′), 4.12 (3H, s, OCH3-9), 4.02 (3H, s, OCH3-10), 3.92 (3H, s, OCH3-7″), 3.18 (3H, t, *J* = 5.1 Hz, H-5). ^13^C-RMN (DMSO-d_6_) δ ppm: 163.8 (C-7″), 161.7 (C-2″), 157.5 (C-9″), 150.3 (C-10), 149.3 (C-3), 146.5 (C-2), 145.6 (C8), 144.2 (C-9), 143.5 (C-2′), 137.4 (C-13a), 134.2 (C4a), 132.6 (C12a), 129.5 (C-5′), 128.8 (C-3′), 127.1 (C-4′), 126.4 (C-5″), 126.1 (C-11), 121.6 (C-12), 121.3 (C-8a), 119.9 (C-13b), 117.1 (C-10″), 114.7 (C-6″), 108.6 (C-4), 107.9 (C-1), 107.6 (C-3″), 102.1 (-OCH2O-), 101.1 (C-8″), 62.2 (OCH3-C-9), 57.1 (C-6), 56.9 (OCH3-C-10), 56.2 (OCH3-C-7″), 35.7 (C-1′), 27.5 (C-5). * (C13) et (C-4″) not observed. Anal. calcd. for C_37_H_30_BrNO_7_: C, 65.30; H, 4.44; N, 2.06. Found: C, 65.20; H, 4.49; N, 1.99.

### 2.2. Biology

#### 2.2.1. Microorganism Strains Used and Growth Conditions

Reference strains used in the study were obtained either from the American Type Culture Collection (ATCC, Molsheim Cedex France), the German Leibniz Institute **(**DSMZ, Braunschweig, Germany) or the French Pasteur Institute (CIP, Paris, France). Environmental and pathogenic Gram-negative bacterial strains used were: *Acinetobacter baumannii* (CIP 110431), *Citrobacter farmeri* (ATCC 51633), *Citrobacter rodentium* (ATCC 51116), *Escherichia coli* (ATCC 8739), *Helicobacter pylori* (ATCC 43504), *Klebsiella pneumoniae* (DSMZ 26371), *Pseudomonas aeruginosa* (ATCC 9027), fluoroquinolone- and carbapenem-resistant *Pseudomonas aeruginosa* (CIP 107398), *Salmonella enterica* (CIP 80.39), *Shigella flexneri* (ATCC 12022), *Vibrio alginolyticus* (DSM 2171) and *Vibrio diabolicus* (from Dr Aurélie Tasiemski, Institut Pasteur Lille). Environmental and pathogenic Gram-positive bacterial strains used were: *Arthrobacter gandavensis* (DSM 2447), *Bacillus subtilis* (ATCC 6633), nisin-resistant *B. subtilis* (DSMZ 347), *Clostridium perfringens* (ATCC 13124), *Clostridium difficile* (DSM 1296), vancomycin-resistant *Enterococcus faecalis* (DSMZ 13591), *Lactococcus lactis* (DSM 20481), *Staphylococcus aureus* (ATCC 6538P), methicillin-resistant *S. aureus* strain MRSA USA300 (ATCC BAA-1717 USA 300 CA-MRSA) and *Streptococcus pyogenes* (DSM 20565). *Mycobacterium smegmatis* (mc2155, ATCC 700084) was used as model of *Mycobacterium* species. Bacteria were cultured as previously described [15,16,17,18] Briefly, most of the bacterial strains were routinely grown on Luria Bertani (LB) agar plates and LB broth at 37 °C in aerobic condition except *H. pylori* that was grown in BHI in micro-aerobic condition using microaerobic BD GasPak generator (Sigma-Aldrich, Lyon, France). *M. smegmatis* was cultured in Middlebrook 7H10 agar plate and Middlebrook 7H10 broth at 37 °C in aerobic condition. *E. faecalis*, *C. difficile*, *C. perfringens* and *S. pyogenes* were cultured in Brain Heart Infusion (BHI) agar plates and BHI broth at 37 °C in anaerobic chamber (Coy Laboratory Products, Grass Lake, MI, USA). Fungal strain *Candida albicans* (DSM 10697) was grown on potato dextrose (PD) agar plates.

#### 2.2.2. Antimicrobial Activity Assay

Antimicrobial activity of berberine and its derivatives was evaluated by determination of their minimal inhibitory concentration (MIC) using two-fold serial dilutions in bacterial liquid media following the National Committee of Clinical Laboratory Standards (NCCLS, 1997) as previously described [15,16,17,18]. Briefly, for most of the bacteria, single colonies of the different bacterial strains cultured on specific agar plates were used to inoculate 3 mL of Mueller-Hinton (MH), except *C. difficile*, *C. perfringens*, *H. pylori*, *E. faecalis*, and *S. pyogenes* that were cultured in 3 mL of BHI and or *M. smegmatis* that was cultured in 3 mL of Middlebrook 7H10 broth. Tubes were then incubated overnight (for approximately 16 h) at 37 °C under stirring (200 rpm), except for *M. smegmatis* that was left to grow for 48–72 h. Optical density (OD) of the bacterial suspensions were then read at 600 nm, adjusted to 1 with medium before bacteria were diluted 1/100 in 3 mL of fresh medium and incubated at 37 °C, 200 rpm until bacteria reached log phase growth (OD_600nm_ around 0.6). In the case of *C. difficile*, *C. perfringens*, *E. faecalis*, *H. pylori*, *S. pyogenes* and *M. smegmatis*, MIC were performed directly using over-night growing suspension. In all cases, bacteria were diluted in appropriate medium to reach bacterial density around 10^5^ cells/mL. 100 μL per well of bacterial suspension were then added into sterile polypropylene 96 well microplates (Greiner BioOne, Dominique Dutscher, Brumath, France). Bacteria were exposed to increasing concentrations of berberine or berberine’s derivatives obtained by serial dilution (from 0 to 100 µM, 1:2 dilution). Stock solutions of berberine and berberine’s derivatives at 10 mM were prepared in DMSO. Volume of DMSO corresponding to the highest dose of berberine’s derivatives tested (1% DMSO final concentration) was used as negative control and was found inactive. Plates were incubated at 37 °C for 18–24 h for all bacteria except *M. smegmatis* that where incubated for 72 h. All bacterial strains were tested in aerobic conditions except *C. difficile*, *C. perfringens*, *E. faecalis* and *S. pyogenes* that were incubated in an anaerobic chamber (Coy Laboratory Products, Grass Lake, MI, USA). MIC on *H. pylori* was performed in a microaerobic atmosphere generated using anaerobic BD GasPak system (Sigma-Aldrich, Lyon, France). For *C. albicans*, liquid suspension was prepared by resuspending colonies collected from LB plates in sterile NaCl 0.85% solution. *C. albicans* were then diluted at 1–2 × 10^3^ cells/mL in RMPI media buffered with MOPS at final concentration of 0.165 mol/L (pH 7.0) before being added to the 96-wells plates in the presence of increasing concentrations of test compounds. Plates were then incubated at 35 °C during 24 h before reading. At the end of the incubation, OD_600nm_ was measured using microplate reader (Biotek, Synergy Mx, Colmar, France). The MIC was defined as the lowest concentration of drug that inhibited visible growth of the organism. Experiments were conducted in independent triplicate (n = 3). Antimicrobial activity of berberine and berberine’s derivatives was also evaluated by determination of the minimal bactericidal concentration (MBC), i.e., the minimum concentration of the test compound killing 99.9% of the bacteria as previously described [15]. MBCs were measured by streaking 10 µL of wells with no observed growth of previously prepared MIC plates on agar plates. After incubation in the proper condition, numbers of colonies were counted. Concentrations where ≤ 1 colony grew were considered to be the MBC.

#### 2.2.3. Cytotoxic Assay on Human Cells

The toxicity of berberine and berberine’s derivatives on human cells was evaluated using resazurin assay as previously described [15,16,17,19,20,21]. Human cells used were BEAS (ATCC CRL-9609), Caco2 (ATCC HTB-37) and HepG2 (ATCC HB-8065) corresponding respectively to human airway epithelial cells, intestinal epithelial cells and liver cells. Cells were cultured in Dulbecco’s modified essential medium (DMEM) supplemented with 10% fetal bovine serum (FBS), 1% L-glutamine and 1% antibiotics (Thermo Fisher Scientific, Illkirch-Graffenstaden, France). Cells were routinely grown on 25 cm^2^ flasks and maintained in a 5% CO_2_ incubator at 37 °C. For toxicity assay, cells were detached using trypsin–EDTA solution (Thermo Fisher Scientific), counted using Mallasez counting chamber and seeded into 96-well cell culture plates (Greiner bio-One, Dominique Dutscher, Brumath, France) at approximately 10,000 cells per well. The cells were left to grow for 48–72 h at 37 °C in a 5% CO_2_ incubator until they reached confluence. Media from wells was then aspirated and cells were treated with 100 µL of culture media containing increasing concentrations of berberine or berberine’s derivatives obtained by serial dilution (from 0 to 50 µM, 1:2 dilution), DMSO (0.5% DMSO final concentration) was used as negative control. After 48 h incubation at 37 °C in a 5% CO_2_ incubator, cell viability was evaluated using resazurin based in vitro toxicity assay kit (Sigma-Aldrich) following manufacturer’s instructions. Briefly, cell wells were emptied, and cells were treated with 100 µL of resazurin diluted 1:10 in sterile PBS containing calcium and magnesium (PBS^++^, pH 7.4). After 4 h incubation at 37 °C, fluorescence intensity (excitation wavelength of 530 nm/emission wavelength of 590 nm) was measured using Synergy Mx microplate reader (Biotek). The fluorescence values were normalized by the controls (DMSO treated cells) and expressed as percent viability. The IC_50_ values of berberine and berberine’s derivatives on cell viability (i.e., the concentration of derivative causing a reduction of 50% of the cell viability) were calculated using GraphPad^®^ Prism 7 software (San Diego, CA, USA). *t*-Test and two ways ANOVA analyses were used to address the significant differences between mean values with significance set at *p* < 0.05.

#### 2.2.4. Bacterial Membrane Permeabilization Assay

Membrane permeabilization by berberine derivatives was evaluated using the cell-impermeable DNA/RNA probe propidium iodide as previously explained [15,16,22,23]. Logarithmic growing bacterial suspensions were prepared from overnight bacterial suspension by 1 in 10 dilution. After 3 h incubation at 37 °C, 200 rpm, bacterial suspensions were centrifuged for 5 min at 3000 rpm. Cell pellets were then resuspended in sterile PBS at about 10^9^ cells/mL. Propidium iodide (Sigma Aldrich) was then added to the suspension at a final concentration of 60 µM. 100 µL of this suspension were then transferred into 96-well black plates (Dominique Dutscher) and exposed to increasing concentrations of berberine’s derivative (from 0 to 100 µM, 1:2 dilution). Volume of DMSO corresponding to 100 µM of derivative was used as negative control and was found inactive. All steps of the assay were performed in aerobic condition except for *H. pylori* which was done in microaerobic condition generated with GasPak unit (Sigma Aldrich). Bacteria were incubated at 37 °C and the kinetics of fluorescence variations (excitation at 530 nm and emission at 590 nm) were then recorded over time using a Synergy Mx microplate reader (Biotek). Cetyl trimethylammonium bromide (CTAB) at 300 µM was used as positive control giving 100% permeabilization. Results were expressed in % of permeabilization. All experiments were done in triplicate.

#### 2.2.5. DNA Fragmentation Assay

Overnight grown bacteria were pelleted at 6000 rpm for 10 min. Bacterial pellets were resuspended in 577 µL of TE buffer (Tris 50 mM, EDTA 50 mM). 30 µL of SDS (at 10%) and 3 µL of proteinase-K (at 20 mg/mL) were then added to lyse the cells. After 1 h incubation at 37 °C, 180 µL of NaCl 5 M were added and tubes were vortexed for 30 s. Chloroform/isoamyl alcohol (24/1, *v*/*v*) was then added. Tubes were vortexed for 30 s and centrifuged at 6000 rpm for 5 min. The supernatants containing extract DNA were then transferred to new tubes containing 1/1 volume of phenol/chloroform/isoamyl alcohol (25/24/1, *v*/*v*/*v*). After vortexing for 5 min and centrifuging at 6000 rpm for 5 min, the supernatants were transferred to new tubes containing 0.6 mL of isopropanol. Tubes were mixed gently until a white DNA precipitate was obtained. Then, the tubes were centrifuged at 10,000 rpm for 10 min. The supernatants were discarded, and the pellets were resuspended in 100 µL of 70% ethanol. The tubes were centrifuged for 5 min at 10,000 rpm and let to dry. Finally, the DNA pellets were resuspended in 50 μL of TE buffer. DNA concentrations were determined at 260 nm and DNA solutions obtained from different bacteria were all adjusted to a concentration of 170 ng/µL. Then 20 µL of the DNA solution were transferred to tubes and were treated with berberine’s derivatives at a final concentration corresponding to 4-times their MIC. After 6 h of incubation at 37 °C, DNA were run onto a 1% agarose gels before being stained using sybr safe and being observed using an Uvitec Cambridge gel viewer (Cambridge, England, United Kingdom). Incubation of DNA for 6 h with sulfuric acid at 10% was used as positive control of fragmentation.

#### 2.2.6. Docking Studies

The crystal structure of FtsZ protein from *S. aureus* in complex with PC190723 inhibitor was selected for the molecular docking studies [24]. The corresponding PDB file 4DXD was prepared using MOE version 2016 (http://chemcomp.com). The binding site was defined as all residues with at least one atom within 14 Å radius from the PC190723 inhibitor. The studied compounds were also prepared using MOE in order to both generate 3D conformers and compute required partial charges. PLANTS was used as the docking engine with its “chemplp” scoring function to generate and evaluate the poses [25]. Its docking algorithm is based on a class of stochastic optimization algorithms called ant colony optimization (ACO). This kind of algorithm, which mimics the behavior of ants finding the shortest path between food and their nest, can be used to efficiently sample the conformational space for docking purpose. Docking simulations of reference ligand PC190723 into its FtsZ structure (PDB code: 4DXD) were beforehand performed to validate the docking protocol. This control study is used to determine the best docking parameters for the target [26]. The PC190723 ligand was successfully redocked into its binding site with a RMSD value of less than 1 Å between crystallized conformation and predicted pose (Appendix A). All generated poses were subjected to visual analysis using Pymol (http://pymol.org) and the binding mode of interest was selected accordingly. The latter should highlight shared binding mode for the berberine core (5 fused-rings substructure) for all considered compounds (**15**, **16**, **24**, **25**) and be reasonable in terms of docking score and explicit interactions with the binding site.

#### 2.2.7. Molecular Dynamics Simulations

The selected binding modes from molecular docking were subsequently subjected to molecular dynamics simulations as final refinement stage. All MD simulations were performed in explicit solvent with periodic conditions with the AMBER package version 18. The initial protein coordinates were taken from the ligand-free crystal structure of FtsZ protein from *S. aureus* (PDB code: 4DXD). MD simulations were performed for each ligand of interest: compounds **15**, **16**, **24**, **25** and the native ligand from the 4DXD structure considered as positive control. In contrast to native ligand, the initial coordinates for compounds **15**, **16**, **24**, **25** were taken from previous docking simulations. The CHARMM-GUI web services (http://www.charmm-gui.org) were used to read the PDB structures and generate inputs and scripts to run the MD simulations [27,28]. For each ligand, the topology and parameter files were generated starting from an external SDF file with the CHARMM General Force Field method. The system was solvated with a pre-equilibrated solvation box of TIP3P water molecules around the protein. Chloride and potassium ions were then added to neutralize the system. Finally, periodic boundary conditions were automatically applied based on the system shape and size. For each considered system, unfavorable contacts were first removed by a short energy minimization protocol (max 5000 minimization cycles), followed by equilibration and production simulations. Equilibration stage consisted of 1 ns simulation (time step 1 fs) with enabled constraints on protein and ligands. Production stage consisted of 50 ns simulation (time step 2 fs) without any constraints. Snapshots of the coordinates were saved every 5000 steps (10 ps) leading to 5000 instantaneous conformations for each trajectory. Trajectories were analyzed using the VMD tool using both built-in and in house Tcl scripts. The main considered criterion was the RMSD of both protein backbone and ligands over the trajectory, after initial superimposition of the CA atoms of the protein in the same referential.

#### 2.2.8. Evaluation of the Effect of Berberine’s Derivatives on the Synthesis of Macromolecules by Bacteria Using Fluorescent Microscopy

Microscopic observation was used to evaluate the effect of berberine’s derivatives on the synthesis of macromolecules by bacteria as previously described [15,16]. Briefly, overnight cultures of bacteria were diluted 1:100 in culture media and were grown at 37 °C with shaking at 200 rpm until they reached an OD_600nm_ of about 0.2. Conventional antibiotics known to inhibit macromolecule’s synthesis (i.e., amoxicillin, gemifloxacin, rifampicin and tetracycline) or berberine’s derivatives were then added at a concentration equivalent to 4-times their MIC and bacteria were incubated at 37 °C for 2 h. At the end of the incubation, bacteria were stained for 15 min on ice for membrane and DNA using 12 μg/mL FM4-64FX (Thermo Fisher Scientific) and 2 μg/mL DAPI (Sigma-Aldrich), respectively. After staining, bacteria were pelleted by centrifugation (centrifugation for 30 s at 7500 rpm) and resuspended in sterile PBS. Bacteria were pelleted a second time and resuspended in 500 µL of sterile PBS. Bacteria were then pelleted a last time and fixed with PBS containing paraformaldehyde (PFA) at 4% (*v*/*v*) for 15 min on ice in the dark. After centrifugation for 1 min at 7500 rpm, the samples were resuspended in 50 µL of mounting solution (Vector Laboratories, Peterborough, UK). Eight µL were transferred onto microscope glass slides and bacteria were observed and images collected using an IX71 Fluoview confocal microscope (Olympus, Rungis, France) with excitation at 543 nm and 405 nm for FM4-64FX and DAPI, respectively.

## 3. Results

### 3.1. Synthesis

The synthetic route of this new series of 13-substituted berberines is outlined in Scheme 1, starting from commercially available berberine chloride (BBR) converted to its 8-acetonylberberine (compound **3**) with NaOH 5M and acetone in 83% yield [14]. Compound **3** was then treated with various bromides and NaI in CH_3_CN yielding 13-substituted berberine derivatives **4**–**24**.

Using 4-formyl berberine (compound **12**) as starting material, we also obtained berberine bearing a sulfonamide moiety in two steps (compounds **22** and **23**). Construction of new dimeric chalcone-berberine scaffold (compounds **24** and **25**) was done by aldol condensation using compound **12** as starting material. Intramolecular cyclization of compound **25** of its chalcone moiety into flavone lead to a new scaffold flavone-berberine (compound **26**) (Scheme 2). NMR spectra of the synthetized derivatives are shown in the Appendix A.

### 3.2. Antibacterial Activity

Antimicrobial activity of the twenty-six different berberine’s derivatives produced in this study was compared to the activity of the original berberine molecule through a first screening assay performed on selected relevant microorganisms representative of Gram-negative (*E. coli* and *P. aeruginosa*), Gram-positive (*B. subtilis* and *S. aureus*), *Mycobacterium* (*M. smegmatis*) and fungi (*Candida albicans*) (Table 1).

As previously published [29], results showed that the natural molecule berberine possesses low to no antimicrobial activity. Oppositely, among the 26 synthetized berberine’s derivatives, many possess higher antimicrobial activity compared to berberine (i.e compounds **2**, **5**, **6**, **8**, **9**, **11**, **13**–**19**, **21** and **23**–**26**). This first antimicrobial activity screening revealed that tested berberine’s derivatives were particularly active against Gram-positive strains (*B. subtilis* and *S. aureus*) with MIC values as low as 1.5 µM for compound **19** for example. *M. smegmatis*, a model of *Mycobacterium*, was also found sensitive to berberine’s derivatives, the lowest MIC value being of 6.25 µM for compound **9**. Some derivatives were also found active against *C. albicans*, the lowest value of MIC being 6.25 µM for compound **19**. Importantly, none of the tested derivatives were found active on the two Gram-negative strains tested (*E. coli* and *P. aeruginosa*). We selected the best derivatives, based on their high antimicrobial activity (lower MIC value) and/or larger spectrum of activity (i.e., active against different microorganisms tested in Table 1) to further study them. The selected derivatives were compounds **2**, **6**, **8**, **9**, **11**, **13**, **14**, **16**–**19**, **23**, **25** and **26**.

### 3.3. Toxicity Evaluation

The next step was to evaluate the toxicity of these selected derivatives using human cells and to compare it to the original berberine molecule (Table 2).

Results showed that toxicity of berberine’s derivatives is cell-type dependent. Thus, in human intestinal epithelial cells (Caco-2 cells), compounds **18**, **19** and **25** were the only ones found more toxic than berberine (i.e., giving lower IC_50_ values) all other compounds being less toxic than the original molecule. For human airway epithelial cells (BEAS cells), in addition to compounds **18**, **19** and **25**, compounds **8**, **9** and **13** were also found more toxic than berberine whereas other derivatives were found less toxic. Finally, in human liver epithelial cells (HepG2 cells), in addition to compounds **9**, **13**, **18**, **19** and **25**, compounds **2**, **6** and **16** were also found more toxic than berberine. Based on good antimicrobial activity and low toxicity, compounds **8**, **16**, **17**, **23**, **25** and **26** were selected and further studied.

The therapeutic indexes (TI) of selected derivatives were calculated by dividing their IC_50_ value on a specific human cell type by their MIC values found on *S. aureus* that gave the lower MIC values (Figure 2 and Table 3).

TI were found dependent on both the specific compound and human cells considered. Thus, for human intestinal epithelial cells (Caco-2 cells), TI ranged from 3.37 to 20.24 with the following order of safety: compound **26** > **23** > **8** > **16** > **17** > **25**. For human airway epithelial cells (BEAS cells), TI ranged from 2.83 to 41.52 with the following order of safety: compound **26** > **16** > **23** > **17** > **8** > **25**. Finally, for human liver epithelial cells (HepG2 cells), TI ranged from 39.39 to 342.00 with the following order of safety: compound **17** > **8** > **23** > **26** > **16** > **25**. Overall, these data showed that although antimicrobial berberine’s derivatives gave good TI values, suggesting that they could potentially be used to treat systemic infections without taking the risk of toxicity for the patients.

Compounds **8**, **16**, **17**, **23**, **25** and **26** were further tested using a large panel of Gram-negative and Gram-positive environmental and pathogenic bacteria, including resistant strains, listed in Table 4. Regarding Gram-positive bacteria, results showed that selected compounds possess good antimicrobial activity (i.e., MIC of 3.12–6.25 µM) against important human pathogens such as *B. cereus*, *S. aureus* or *S. pyogenes*. Other Gram-positive pathogenic bacteria, such as *C. difficile*, *C. perfringens* or *E. faecalis* were found less sensitive giving MIC values superior or equal to 12.5 µM. Importantly, Gram-positive strains resistant to conventional antibiotics, i.e., nisin-resistant *B. subtilis*, methicillin-resistant *S. aureus* and vancomycin-resistant *E. faecalis*, were found sensitive to berberine’s derivatives. Interestingly, results confirmed that derivatives were inactive on most of the tested Gram-negative bacteria, except *H. pylori* and *V. alginolyticus* that were found sensitive with MIC values as low as 1.5–3.12 µM. Determination of the Minimal Bactericidal Concentration (MBC) of the selected compounds showed that their MBC values were closed or identical to their MIC values (Appendix A), indicating that they are bactericidal rather than bacteriostatic.

### 3.4. Mechanistic Analysis

Compounds **16** and **25** showing the higher activity on a large number of bacteria, they were further investigated in term of mechanism of action using different approaches, their structurally-related inactive derivatives (i.e., compounds **15** and **24**) being used as controls (Figure 3). 

#### 3.4.1. Membrane Permeabilisation Assay

Previous studies have shown that some berberine’s derivatives have the ability to cause membrane permeabilization [15,16,22]. For this reason, the effect of the compounds **15**, **16**, **24** and **25** on the bacterial membrane integrity was investigated (Figure 4). As expected, inactive derivatives (i.e., compounds **15** and **24**) caused low to no membrane permeabilization (data not shown). Active derivatives (i.e., compounds **16** and **25**) caused membrane permeabilization, with the following order of efficiency: *V. alginolyticus* = *S. aureus* > *H. pylori* > *B. subtilis* with a maximal permeabilization of 80–100; 50–60 and 10–15%, respectively after 120 min exposure to 100 µM of derivative. 

In all cases, compound **16** was more active than compound **25**. Importantly, although membrane damages were observed at high doses, at 4-times their MIC compounds **16** and **25** caused low to no membrane permeabilization, except for *V. alginolyticus* where 20–40% permeabilization was observed, demonstrating that the antibacterial effect of these compounds rely on other mechanism(s).

#### 3.4.2. DNA Fragmentation Assay

The ability of compounds **16** and **25** to cause DNA fragmentation was then evaluated, known antibacterial berberine’s derivatives obtained by modification at position 13 with alkyl and diphenyl alkyl possessing such activity [11]. Results showed that incubation of bacterial DNA extracted from *B. subtilis* or *V. alginolyticus* for 6 h at 37 °C with compounds **16** and **25** at 4-times their MIC did not caused DNA fragmentation (Appendix A). Similarly, compounds **16** and **25** failed at causing the fragmentation of DNA extracted from *S. aureus* and *H. pylori* (data not shown) ruling out the implication of such mechanism in their antibacterial effect.

#### 3.4.3. Molecular Docking

FtsZ protein from *S. aureus* (FtsZ, PDB code: 4DXD) has been previously shown, using in vitro biological assays, to be the target of antibacterial berberine’s derivatives [30]. Since FtsZ protein is involved in septum formation and bacterial cell division, the interaction of berberines with this bacterial protein was proposed to explain their antibacterial activity [30]. For this reason, molecular docking studies were conducted on the FtsZ protein from *S. aureus*. The berberine core (5 fused-rings substructure) and compounds **15**, **16**, **24**, **25** were investigated. Based on docking experiments (Figure 5), the core berberine substructure was predicted to be deeply buried in a hydrophobic pocket from the binding site, with an overlap with the known PC190723 inhibitor from reference 4DXD structure.

More precisely, the ABC rings (from the dioxole end) of the berberine core were superimposed on the chlorothiazolopyridine moiety of PC190723. Interestingly, the variable sidechain was also directed toward the exit of the binding site, allowing large modifications as highlighted by the diversity of the sidechains from reported active compounds in this study. While compound **25** was successfully predicted as the best compound (score criterion, favorable contacts with the site) among the five considered structures, the models were not able to explain the observed activity differences between close analog pairs (**15**/**16** and **24**/**25**). The potent compound **25** is expected to create additional favorable contacts with the binding site, especially a hydrogen bond between its hydroxyl group and the polar sidechain from E305 (Figure 5). At this stage, docking analysis suggested that at least compound **25** is able to interact with FtsZ, a protein involved in bacterial division. It should be noted that this predicted binding mode for the berberine core was previously proposed for the cycloberberine analog series [13]. These predicted binding modes from docking were then subjected to molecular dynamics simulations as final refinement (Figure 6). The monitoring of RMSD criterion over the trajectory clearly highlighted that the docking binding modes were not reliable, as the complexes were not stable starting from the beginning of the production trajectories. As a control, the same molecular dynamics simulations were performed on the original 4DXD complex and small RMSD values were obtained in this case in contrast to the berberine series.

#### 3.4.4. Macromolecule Synthesis Inhibition Assay

Since the antibacterial activity of compounds **16** and **25** does not rely on membrane permeabilization, DNA fragmentation or FtsZ interaction (i.e., the reported mechanisms of action of already known berberine derivatives) their effect on bacterial synthesis of macromolecules was finally evaluated using fluorescent microscopy as previously described (Table 5 and Table 6. Figure 7, Figure 8, Figure 9, Figure 10, Figure 11 and Figure 12) [16,17].

This technique based on the specific fluorescent labelling of bacterial followed by microscopic observation represents an alternative to radioactivity and was successfully used by others and us to evaluate the effect of various antibacterial agents on macromolecule synthesis [16,17]. The assay relies on the comparison of the morphological changes caused by test compounds and conventional antibiotics with known mechanism of action used as controls. The assay was performed on *B. subtilis* (Figure 7, Figure 8 and Figure 9) and *V. alginolyticus* (Figure 10, Figure 11 and Figure 12) as models of Gram-positive and Gram-negative sensitive strains. Bacteria were exposed to compounds **16**, **25** or conventional antibiotics at 4-times their MIC values for 2 h before labelling of their membrane using FM 4–64 (red fluorescence) or their DNA using DAPI (blue fluorescence) and microscopic observation. Results showed that compound **16** caused morphological changes in *B. subtilis* (Table 5) and *V. alginolyticus* (Table 6) similar to the ones caused by amoxicillin known to inhibit cell wall synthesis, whereas compound **25** caused bacterial morphological changes similar to the ones observed with rifampicin known to inhibit the synthesis of RNA.

## 4. Discussion

A series of 13-substituted berberine derivatives were synthesized and evaluated in term of antimicrobial activity, toxicity, and mechanism of action. Evaluation of the antimicrobial activity and toxicity of the derivatives showed that biological activities were affected by the substituent present at position 13. Analysis of the structure-activity relationship showed that with the simple *para* substitution in compounds **6**–**12**, only compound **8** possessing an iodomethyl group and compound **9** with an allyl group had interesting bactericidal activities. Unfortunately, compounds **8** and **9** were also found more toxic than berberine. Regarding compounds **13**–**17** which are multi-substituted on the phenyl ring, only compound **15** (3,4,5 trimethoxy phenyl) was found inactive. Replacement of the 4-methoxy group of compound **15** by a 4-benzoate group (as in compound **16**) or with benzoate at position 2 and 4 (as in compound **17**) increased the antimicrobial activity. Increasing the size of substituent on the phenyl ring seems to enhance the antimicrobial activity without increasing the toxicity of the compound. The presence of a coumarin (as in compound **18**) or benzothiazole (as in compound **19**) dramatically increased the antimicrobial activity but also the toxicity of the molecules. Introducing a sulfonamide (in compound **23**) gave activity comparable to that of compound **22** and reduced toxicity compare to berberine. The presence of a 4-acetamido chalcone at position 4 (as in compound **24)** leaded to an inactive molecule, whereas the presence of a 5-methoxy chalcone at the same positon (as in compound **25)** led to an active form. Finally, compound **26** corresponding to the first in class flavones-berberine was found less active than compound **25**. The antimicrobial activity of the berberine’s derivatives described in the present study was meanly directed against Gram-positive bacteria, including the ones part of the high priority group of the WHO’s list such as *Staphylococcus*, *Enterococcus* and *Streptococcus*, but also against *Mycobacterium* and *C. albicans*. Gram-negative bacteria were not sensitive to the berberine derivatives except *V. alginolyticus* and *H. pylori* that were found sensitive to compounds **8**, **16**, **17**, **23**-**26**. The antibacterial action of these compounds on *H. pylori* is very interesting since this bacteria is classified as priority pathogen by the WHO due to its prevalence (i.e., 50% of the adults worldwide and even 80–90% of adults in developing countries, making it the second cause of bacterial infection after caries) and its severe complications (i.e., gastric ulcers in 10% of infected patients and gastric cancer in 1% of the patients that is estimated to kill 700,000 persons per year) [31]. Mechanistic approaches were performed for compounds **16** (3,5-dimethoxy-4-benzoate phenyl derivative) and **25** (5-methoxychalcone derivative) demonstrated that, contrary to previously described 13-substituted berberine derivatives, these derivatives do not cause membrane permeabilization or DNA fragmentation. Docking studies were used to predict the binding mode of several compounds from this series including the potent compound **25**. The global binding mode, with the berberine moiety deeply buried within the binding site of FtsZ, was previously described for an analog cycloberberine series [13]. However, refinement of these predicted binding modes using molecular dynamics simulations suggested that they were not reliable. Given this and the fact that the activity cliffs could not be explained by the models (compounds **15** vs. **16;** compounds **24** vs. **25**), the latter were not validated for this new series of compounds. Taken together mechanistic studies showed that the new active berberine’s derivatives described in this study are not acting through mechanisms of action reported for already known berberine’s derivatives (i.e., membrane permeabilization, DNA fragmentation or FtsZ interaction). Morphological microscopic approach demonstrated that compounds **16** and **25** have an original mechanism of action for berberine’s derivatives relying on their ability to inhibit the synthesis of the cell wall and of RNA, respectively.

## 5. Conclusions

In conclusion, the good activity of the 13-substituted berberine derivatives developed in the present study against pathogenic bacteria, including resistant strains, associated to their low toxicity against human cells, suggest that these molecules should be further evaluated and considered to treat bacterial infection, particularly in the context of world-wild increase in antibiotic resistance of bacteria.

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
