# Peer review of "Synthesis and Evaluation of the Antibacterial Activities of 13-Substituted Berberine Derivatives"

_antibiotics, 2020, doi:10.3390/antibiotics9070381_

Round 1

Reviewer 1 Report

Dear Authors, 

All of the comments and suggestions are included in the PDF file. 

Despite the fact, that the manusript is relativlely interesting, it should be improved before the publication (ommisions, editiorial errors ect.).

All best,

Author Response

Reviewer 1:

We would like to thank Reviewer 1 for all his/her constructive and helpful comments / suggestions that allow us to improve our manuscript.

We checked the full document provided by Reviewer 1 and performed corrections / modifications accordingly. Please see directly the revised manuscript where all changes are indicated.

More specifically:

  • Reviewer 1 asked us to provide name, city and country for all companies mentioned in the manuscript. Answer: This has been done in revised manuscript.

  • Reviewer 1 asked us to move the sentence line 386 to the Acknowledgments. Answer: This has been done in revised manuscript.

  • Reviewer 1 pointed the fact that Lactococcus lactis is not a pathogenic strain. He/She is right. Answer: We indicated in the revised manuscript that Gram-negative and Gram-positive bacteria tested are either environmental or pathogenic strains (Lines 389 and 395).

  • Reviewer 1 asked what ISO norm was used to determine the MBC (Lines 434-437). Answer: We thanks Reviewer 1 for his/her question. We did not used a specific ISO norm to determine MBC. We used a procedure used by others and us to determine MBC from the over-night MIC plates. He procedure is detailed in the revised manuscript This technic does not correspond to any ISO norms but it has been validated and used in numerous publications to evaluate if the antibacterial effect of a molecule is due to an bacteriostatic or bacteriolytic effect and to determine MBC values.

  • Reviewer 1 suggested to add the fluorescence microscopy pictures to the main text and to remove Tables 5 and 6. Answer : As suggested by Reviewer 1, we moved the microscopy data from the supplementary part to the main text (new Figures 6-11). But we truly believe that the Tables will help the readers to analyze those pictures and we would like to ask the Reviewer and Editor the authorization to maintain the Tables in the revised manuscript.

We hope that our answers will satisfy the reviewer and that our revised manuscript will be considered for publication in the Antibiotics.

Sincerely yours,
Dr Maxime Robin and Marc Maresca

Reviewer 2 Report

Before docking.... no inhibitory tests were performed against selected target, e.g.: Surface Plasmon Resonance (SPR) or Localized Surface Plasmon Resonance (LSPR), or Isothermal Titration Calorimetry (ITC), or Microscale Thermophoresis (MST) or Biolayer Interferometry (BLI).

What is “purpose.edocking” (line 507)?

The mechanism of action and docking (lines 509-516, Supplementary Material Image S1, 3.4.2 Molecular Docking) were treated very lightly. Personally, I consider “visual analysis” of docking poses an unfortunate way to summarise.

According to DrugBank.ca, the target of Amphotericin B is Ergosterol, respectively, the targets of Gemifloxacin are DNA topoisomerase 4 subunit A & DNA gyrase subunit A... therefore, how authors can correlate the results of  docking performed against Cell division protein FtsZ with evaluation of antimicrobial activity (3.2 Antibacterial activity  and Table 1)??? There are different mechanisms of actions. MD simulations where not performed as post-docking operations to show that the protein-ligand complexes are stable in time.  Section 3.4.3. Macromolecule synthesis inhibition assay, can’t explain why FtsZ was chosen as target, moreover no BLAST was performed to verify if the structure of FtsZ is highly conserved at all species investigated in this paper.

Author Response

Reviewer 2:

We would like to thank Reviewer 2 for all his/her valuable and constructive comments / suggestions that allow us to improve our manuscript.

  • Reviewer 2 pointed the fact that no inhibitory tests were performed against selected target, e.g.: Surface Plasmon Resonance (SPR) or Localized Surface Plasmon Resonance (LSPR), or Isothermal Titration Calorimetry (ITC), or Microscale Thermophoresis (MST) or Biolayer Interferometry (BLI) prior to do the docking analysis. Answer: Since FtsZ protein is an already characterized and published target of berberine’s derivatives with antibacterial effect, we performed docking analysis of our derivatives with FtsZ without the need to confirm their ability to interact with FtsZ through other techniques. The fact that FtsZ was previously identify as a target of berberine’s derivatives allowing docking analysis to be used as been clearly stated in the revised manuscript (Lines 707-711). The fact that the docking analysis is able only to predict (and not to confirm) the interaction of our derivatives with FtsZ protein is also mentioned in the revised manuscript (Lines 25 and 723-724). The physical interaction of our derivatives with FtsZ protein may be done in a future study more focusing on FtsZ interaction. Nevertheless, we believe that the scope of the present study focuses more the description of these new berberine’s derivatives, the evaluation of their antimicrobial activity and innocuity and a preliminary investigation of their mechanism of action based on mechanisms of action of already characterized berberine’s derivatives such as DNA fragmentation, membrane permeabilisation or FtsZ interaction. In addition to these already identified / described mechanisms of action, we also evaluated the ability of our derivatives to inhibit macromolecules synthesis, something not evaluated yet with berberine and its derivatives, adding novelty to our study compared to existing ones.

  • Reviewer 2 pointed an error Line 507 (also pointed by Reviewer 1: Answer : This error has been corrected in the revised version of our work.

  • Reviewer 2 pointed that the mechanism of action and docking (lines 509-516, Supplementary Material Image S1, 3.4.2 Molecular Docking) were treated very lightly. Personally, I consider “visual analysis” of docking poses an unfortunate way to summarize. Answer: As explained above, since FtsZ protein is an already characterized and published target of berberine’s derivatives with antibacterial effect, in our opinion the docking analysis of the interaction of our derivatives with FtsZ does not require confirmation of their ability to interact with FtsZ through other techniques in the present study that already contains a lot of data about activity, innocuity and mechanism of action. We are planning to focus on FtsZ interaction in a near future. But due to the situation in France and worldwide, we don’t know when this will be possible. We hope that Reviewer 2 will agree that the present study contains already an important amount of data regarding the description of new berberine’s derivatives, the evaluation of their antimicrobial activity and innocuity and a preliminary investigation of their mechanism(s) of action, through classical techniques but also, originally through an evaluation of their ability to inhibit macromolecules synthesis, something not evaluated yet with berberine and its derivatives, adding novelty to our study compared to existing ones.

  • Reviewer 2 pointed the fact that according to DrugBank.ca, the target of Amphotericin B is Ergosterol, respectively, the targets of Gemifloxacin are DNA topoisomerase 4 subunit A & DNA gyrase subunit A... therefore, how authors can correlate the results of docking performed against Cell division protein FtsZ with evaluation of antimicrobial activity (3.2 Antibacterial activity and Table 1)??? There are different mechanisms of actions. Answer: We thanks Reviewer 2 for this comment. As mentioned above, we investigated the mechanism of action of our derivatives based on mechanisms of action already identify for berberine’s derivatives possessing antimicrobial activity. This includes : i) interaction with FtsZ, ii) membrane permeabilisation, iii) DNA fragmentation. Based on that, we evaluated the effects of our derivatives on each of these already identified mechanisms of action. We found that our derivatives were not able to cause membrane permeabilisation nor DNA fragmentation. But they were able to interact with FtsZ, at least based on docking analysis. In addition to already known mechanisms of action of berberine’s derivatives, we used an innovative technique based on fluorescence microscopy to evaluate the effect of our derivatives on macromolecule synthesis adding novelty to our study compared to existing ones. We agree with Reviewer 2 that there is not direct link between interaction with FtsZ and other mechanisms of action such as macromolecule synthesis inhibition. But we did not mentioned such link in the original or revised manuscript.

  • Reviewer 2 pointed the fact that the MD simulations where not performed as post-docking operations to show that the protein-ligand complexes are stable in time. Answer : We agree with Reviewer 2 that molecular dynamics simulations are often used as post-processing refinement of docking outputs. However, due to the nature of this study mainly focused on experimental results about chemistry synthesis and microbiology, we considered that this kind of post-processing was not necessary. The two goals of the present molecular modeling studies were to propose a potential binding mode for the berberine series and to reproduce (or not) the previously reported binding mode of the cycloberberine series from a recent paper. We grant Reviewer 2 to decide if this part of the manuscript should be removed, if she/he considers that it does not meet the requirements for Antibiotics journal. It should also be noted that any molecular dynamics simulations could not be performed in the short to medium term because of the quarantine situation in France.

  • Reviewer 2 pointed the fact that in Section 3.4.3. Macromolecule synthesis inhibition assay, can’t explain why FtsZ was chosen as target. Answer: We already addressed this point. Please see our answer above.

  • Reviewer 2 asked if BLAST was performed to verify if the structure of FtsZ is highly conserved at all species investigated in this paper. Answer: The goal of protein Blast search is to retrieve proteins with high sequence identity to the query without any bias. Here, multiple sequence alignment of homologous FtsZ proteins from diverse bacteria could be sufficient. However, we did not perform such kind of sequence alignment, mainly because we focused on the FtsZ protein from aureus for two important and reliable reasons. First, we observed potent activity for S aureus in this study, and second, berberine compounds were already described as FtsZ inhibitors using in vitro tests on the FtsZ protein of S. aureus.

We hope that our answers will satisfy the reviewer and that our revised manuscript will be considered for publication in the Antibiotics.

Sincerely yours,
Dr Maxime Robin and Marc Maresca

Round 2

Reviewer 2 Report

The authors tried to dodge almost of all my comments/recommendations, hereby I consider that the article should be rejected.

I can accept that SPR (or similar suggested techniques) shouldn't be performed in the current pandemic context - it is a fair POV. However, I asked  for additional in silico experiments, which can be performed safely from home or remote access or by an (external) expert which can do that for them. Sorry, but the pandemic context it is not an excuse for refusing to perform in silico work and shouldn't be a criteria for publishing - at least not for papers non-related with SARS-CoV-2/COVID-19.

Author Response

  • We sincerely apologie if the Reveiwer was offended. Reviewer 2 pointed the fact that the MD simulations where not performed as post-docking operations to show that the protein-ligand complexes are stable in time. Answer : We sincerely thank Reviewer 2 for urging us to perform molecular dynamics simulations. The results of the MDS showed that Reviewer 2 was right. The monitoring of RMSD criterion over the trajectory clearly highlighted that the docking binding modes were not reliable, as the complexes were not stable starting from the beginning of the production trajectories. As a control, the same molecular dynamics simulations were performed on the original 4DXD complex and small RMSD values were obtained in this case by contrast to the berberine series. We gratefully thank Reviewer 2 for suggesting us to do this analysis that confirm that FtsZ binding is not the mechanism of action of our molecules. The final manuscript has been completely modified accordingly, from the abstract to the conclusion (modified text is underlined in yellow) (Lines 29-32; Lines 532-554; Lines 734-773; Lines 903-914). Mr Taher Yacoub and Dr Laurent Hoffer were involved in the MDS analysis. They did a long analysis so their names were moved closer for the 1rst author name. Mr Taher Yacoub was not initially involved in the study so his name was added to the list of authors of the revised manuscript.

We hope that our answers will satisfy the reviewers and that our revised manuscript will be considered for publication in the Antibiotics.

Round 3

Reviewer 2 Report

Line 527 and Image S1: PC190723 into its FtsZ structure: can be removed. The spatial coordinates of docking site can bu useful to iterate as text.